# Potential Theranostic Roles of SLC4 Molecules in Human Diseases

**DOI:** 10.3390/ijms242015166

**Published:** 2023-10-13

**Authors:** Jingwen Zhong, Jing Dong, Wenyan Ruan, Xiaohong Duan

**Affiliations:** State Key Laboratory of Oral & Maxillofacial Reconstruction and Regeneration, National Clinical Research Center for Oral Disease, Shaanxi Key Laboratory of Stomatology, Department of Oral Biology & Clinic of Oral Rare Diseases and Genetic Diseases, School of Stomatology, The Fourth Military Medical University, Xi’an 710032, China; zhong17770743082@163.com (J.Z.); 17836188070@163.com (J.D.); rwy12345679@163.com (W.R.)

**Keywords:** solute carrier family 4, transporter, physiology, genetic disease, therapy, diagnose

## Abstract

The solute carrier family 4 (SLC4) is an important protein responsible for the transport of various ions across the cell membrane and mediating diverse physiological functions, such as the ion transporting function, protein-to-protein interactions, and molecular transduction. The deficiencies in SLC4 molecules may cause multisystem disease involving, particularly, the respiratory system, digestive, urinary, endocrine, hematopoietic, and central nervous systems. Currently, there are no effective strategies to treat these diseases. SLC4 proteins are also found to contribute to tumorigenesis and development, and some of them are regarded as therapeutic targets in quite a few clinical trials. This indicates that SLC4 proteins have potential clinical prospects. In view of their functional characteristics, there is a critical need to review the specific functions of bicarbonate transporters, their related diseases, and the involved pathological mechanisms. We summarize the diseases caused by the mutations in *SLC4* family genes and briefly introduce the clinical manifestations of these diseases as well as the current treatment strategies. Additionally, we illustrate their roles in terms of the physiology and pathogenesis that has been currently researched, which might be the future therapeutic and diagnostic targets of diseases and a new direction for drug research and development.

## 1. Introduction

Solute carrier family 4 (SLC4) is the major group of transmembrane bicarbonate transporters that mediates bicarbonate secretion in the epithelial cells of multiple organs such as the kidneys, brain, stomach, and intestine. The SLC4 family of transporters includes 10 members in mammals, including three anion exchangers (SLC4A1-3), five Na^+^-coupled HCO3^−^ transporters (SLC4A4-5, SLC4A7-8, SLC4A10) and two other members (SLC4A9 and SLC4A11). The SLC4 family members are widely expressed in various fluid-transporting epithelial and related cells and are responsible for regulating cellular pH, cell volume, and cell signal transduction through transmembrane bicarbonate transporting. The regulating cellular volume of SLC4 molecules plays a crucial role in cellular migration [1] and in osmotic regulation. Mutations in these proteins may explain and impact the phenotype of certain diseases, including cancer. The clinical prospects of the lost activity of certain SLC4 proteins in carcinomas can be utilized for diagnosis and therapeutic applications [1]. Any defects in SLC4 family proteins may lead to changes of multiple biological processes related to pH, including neuronal excitability [2], cardiovascular function [3], the absorption of HCO_3_^−^ in the proximal cells and distal renal tubule [4,5], and pH regulation in the cornea [6]. The mutations in *SLC4* family genes are associated with many diseases, such as hereditary spherocytosis (HS) [7], distal renal tubule acidosis (dRTA) [5], oxidative stress (OS) [8], epilepsy [9], retinal diseases [10], cognitive impairment [11], etc., (Figure 1). These disorders can be diagnosed by clinical examination, genetic screening, and a variety of laboratory tests. In addition, the homeostasis of intracellular pH is often significantly altered in cancer [12] Therefore, SLC4 family proteins have come to be considered therapeutic targets in many cancers. In this review, we will describe the functions of SLC4 family members in different organs and discuss the main pathogenesis of their associated diseases. Table 1 summarizes their expression sites, physiological functions, and pathological processes. Furthermore, it is crucial to investigate potent inhibitors targeting these proteins and develop targeted therapies in the future.

## 2. Topological Structure, Structural Difference of SLC4 Proteins

SLC4 family members are integral membrane proteins and function to transport ions across the membrane, which contain N-terminal glycosylation sites. The structure of SLC4 proteins consists of a large intracellular N-terminal (Nt) region, a multiple-spanning transmembrane domain (TMD), and a small intracellular carboxyl-terminal (Ct) domain [13].

Foremost, topological model analysis showed that SLC4A4 has a large third extracellular loop (EL3) and a small fourth extracellular loop (EL4) [6]. In SLC4A11, the largest extracellular loop is located between TM5 and TM6 [14], while the EL3 in SLC4A1-3 is obviously shorter [15]. SLC4A2 has one cysteine residue at EL3, SLC4A9 has four cysteine residues at EL3 and SLC4A1, SLC4A3, and SLC4A11 have no cysteine residues on their EL3. SLC4A2 has only one cysteine residue on its EL3. SLC4A9 has four cysteine residues located on EL3. Sodium-coupled transporters, including SLC4A4, SLC4A5, SLC4A7, and SLC4A10, have four highly conserved cysteines residues on EL3 [16]. These four cysteine sites are intramolecular disulphide, forming highly ordered topological domains [17]. For sodium-coupled transporters, disulfide bonds at EL3 are various. In SLC4A4, S-S bond formation involves the first and second, and third and fourth cysteine residues, as evidenced by previous biochemical and functional mutagenesis results [18]. The complete glycosylated EL3 loop of SLC4A8 is characterized by a well-folded α/β domain. Located on the domain interface, two disulfide bridges are responsible for stabilizing the α/β domain, whose stability is regulated by two disulfide bridges located at the domain interface. The S-S bonds in SLC4A8 are formed between the first and fourth, and the second and third cysteine residues of EL3, whereas in SLC4A4, the S-S bond formation involves the first and second, and third and fourth cysteine residues. It remains unknown whether this difference of S-S bond formation at EL3 S-S is related to different transportation of SLC4A8 and SLC4A4 [15]. EL3 is also critical for the structure of SLC4A11. It has been reported that three mutants, located in the EL3, p.Val507Ile, p.Val575Met, and p.Tyr526Cys, are associated with the development of Fuchs endothelial corneal dystrophy (FECD), which may be caused by mutations in this region disrupting the structure of SLC4A11 [14].

SLC4A1 is the only member of SLC4 family that contains the N-glycosylation motif “NSSA” at EL4. The extracellular loop, located in the seventh and eighth transmembrane segments, contains an N-terminal glycosylation site known as Asn642. This glycosylation site serves as a marker indicating that the erythrocyte is located outside the membrane [19]. The previous study shows that EL4 is the region that determines protein electrogenicity [20]. EL4 of SLC4A4 is thought to be related to its electrogenic properties. The fourth extracellular loop contains a large number of proline residues, which is also important to the electrogenic properties of the transporters. EL4 can be interact with amino acid residues embedded in the lipid bilayer, altering ionic interactions [17].

In addition, the research has demonstrated that SLC4A1, SLC4A4, and SLC4A8 show significant differences in putative ion coordination regions and permeating cavities. The binding pockets of SLC4A8 and SLC4A4 are similar but differ from that of SLC4A1. The main difference is the presence of a positively charged residue (Arg730) in the protein center of SLC4A1, rather than the nonpolar residue found in Na^+^-dependent SLC4A8 and SLC4A4. Moreover, in the charged residues of TM3 and TM5, there were significant differences between the three proteins [15].

At last, SLC4 proteins mediate distinct ion-transporting mechanisms. It is known that SLC4A11 is not a HCO_3_^−^ transporter, unlike other members of the SLC4 family. Due to the structure-functional trait, there is a notable difference between SLC4A1 and SLC4A11. The functional differences between SLC4A11 and SLC4A1 lie in the presence of a histidine residue in SLC4A11 at position 724 [21]. In the catalytic site, SLC4A11 is the only member of SLC4 family member without an arginine at this position. Additionally, the cytoplasmic domain and membrane domain of SLC4A11 were predicted to be closely associated, while SLC4A1 has an interaction that is not so tight. Of note, SLC4A11 has a highly conserved asparagine-proline-X (NPX) water channel motif (N639PS) in TMD [22].

## 3. Classification and Ion-Transporting Mechanism

According to their functional characteristics, the SLC4 family can be subdivided into three types: (1) Na^+^-independent electroneutral Cl^−^/HCO_3_^−^ exchangers, including SLC4A1-3; (2) Na^+^-HCO_3_^−^ cotransporters, including SLC4A4 and SLC4A5, which are electrogenic, whereas SLC4A7 and SLC4A10 are electroneutral; and (3) Na^+^-coupled HCO_3_^−^ transporter (SLC4A8). The other two members, namely SLC4A9 and SLC4A11, are segregated from the family due to their disparate amino acids and structural and functional characteristics [16]. SLC4 proteins mediate distinct ion-transporting mechanisms. SLC4A1-3 mediates the transmembrane flow of Cl^−^ in exchange for HCO_3_^−^ [16]. SLC4A4 mediates unilateral movement of sodium and bicarbonate across the plasma membrane at a ratio of 1:2 (the “inflow” mode) or 1:3 (the “outflow” mode) [23]. However, recent studies demonstrated that SLC4A4 is capable of combining only two ions, like HCO_3_^−^ or CO_3_^2−^, by using molecular simulation. Thus, SLC4A4 operating in the “inflow” mode moves one Na^+^ and two HCO_3_^−^, whereas SLC4A4 in the “outflow” mode moves one Na^+^, one HCO_3_^−^ and one CO_3_^2−^ [24].

SLC4A5 represents an electrogenic Na^+^-2HCO_3_^−^ or Na^+^-3HCO_3_^−^ cotransporter [19,25]. SLC4A7 absorbs Na^+^ and HCO_3_^−^ electroneutrally. It can plentifully permeate sodium without the transport of bicarbonate [26]. SLC4A10 moves Na^+^ and HCO_3_^−^ unidirectionally [27]. Whether the transport process is followed with the Cl^−^ efflux remains controversial [28]. SLC4A9 is identified as a Cl^−^/HCO_3_^−^ exchanger that can permeate cations nonselectively and transport ions electroneutrally. It is responsible for mediating the influx of Cl^−^, the efflux of 2 HCO_3_^−^, and Na^+^-like monovalent cations, such as Cs^+^, etc. [29]. SLC4A11 is unable to transport HCO_3_^−^. Instead, it mediates electrogenic Na^+^-coupled borate transport [30] and NH_3_/H^+^ cotransport [31] and can serve as an aquaporin [22]. Two models for the activity of SLC4A11 have been put forward, namely the H^+^(OH^−^) conductance and the model of NH_3_-H^+^.

## 4. The Roles of SLC4A Proteins in Human Tissues

### 4.1. Anion Exchangers

SLC4A1 is mainly distributed in erythrocytes and renal cells. In erythrocytes, SLC4A1 is termed as Band 3 [32]. SLC4A1 can not only catalyze bidirectional transport of Cl^−^ and HCO_3_^−^, but also anchor the cytoskeleton, thus maintaining the stability of the erythrocyte membrane. In erythrocytes, SLC4A1 is involved in a significant process of gas exchange [33] (Figure 2), which can regulate pH in the blood (Figure 3). SLC4A1 is also located in the basolateral kidney and is regarded as kAE1 [34]. Renal intercalated-A cells can transport inward HCO_3_^−^ in exchange for outward Cl^−^. Through cooperating with the other proteins, they mediate the transport of H^+^ into the lumen and fulfill the transport of bicarbonate into the interstitium, thus regulating the pH in the blood [35]. SLC4A1 can also be found in the tissue of the epididymis, which is a part of the male reproductive tract [36]. It is reported that SLC4A1 participates in the sperm capacitation process and functions in the rearrangement of the sperm membranes, having a role in the acrosome reaction [37].

SLC4A2 is distributed in the digestive tract [38], respiratory tract [39], nervous and urinary system [40]. Bicarbonate can be secreted by SLC4A2, which is located in the esophageal submucosal glands (SMG). The accumulation of bicarbonate plays a role in buffering acid and protecting epitheliums. It can protect the esophagus from acidic corrosion by neutralizing acidic reflux. In addition, SLC4A2 contributes to the bicarbonate excretion of parietal and mucous cells and serves an important function in providing chloride for the gastric lumen. Through secreting HCO_3_^−^ and excreting Cl^−^ in an electroneutral manner, SLC4A2 is responsible for formulating the liquid, buffering the intraluminal environment within the optimal pH, and preventing the small intestine from absorbing gastrin acid. It is speculated that SLC4A2 can be an alternative pathway that assists the intestine epithelium to take in Cl^−^, which has been observed similarly in submandibular acinar cells. In the pancreas, the acinar cells initially produce the liquid abundant in Cl^−^ and the ductal cells secrete large amounts of pancreas juice and HCO_3_^−^. The reduction of SLC4A2 activities is important for decreasing chloride concentration inside cells and maximizing secreted HCO_3_^−^ concentration. Whether SLC4A2 is involved in secreting HCO_3_^−^ remains unclear. In human cholangiocytes, SLC4A2 represents the major acid-loading mechanism and exhibits the main effect of Cl^−^/HCO_3_^−^ exchange activity. Therefore, its expression functions in the formulation of the biliary HCO_3_^−^ umbrella, which can protect themselves against bile-salt induced injury [38].

SLC4A2 was expressed in monocyte and osteoclasts [41]. During the early stages of osteoclast formation, osteoclasts form actin rings and integrin-based cytoskeletal structures called podosomes [42]. Podosomes are involved in cell adhesion, spreading, and migration. It is regulated by pH-sensitive cysteine proteases. In addition, during osteoclast formation, the podosomes coalesce to form a circumscribed band, forming a sealing zone. When osteoclasts mature, the seal is distinguished into two parts: the surface of the cavity and the absorptive surface [43]. SLC4A2 is significant in mediating intracellular pH and regulating podosome disassembly, which plays a role in osteoclastogenesis. Foremost, SLC4A2 is involved in osteoclast apoptosis and maturation in the dynamic organization of the podocyte [42]. It has been demonstrated that SLC4A2 participates in cytoskeletal organization in osteoclasts through the regulation of calpain activity via controlling intracellular pH [44]. Located on the contra-lacunar surface, SLC4A2 is responsible for exchanging inward HCO_3_^−^ for outward Cl^−^ (Figure 2). Cl^−^ can be transported to the resorption lacuna through the chloride channel to reduce pH_i_, activate pH-sensitive cysteine proteases, then mediate the organization of podosomes in osteoclasts to form actin belts and support cell spreading [42]. SLC4A2 plays an acid-base regulatory role in osteoclasts. The low pH within osteoclasts helps to maintain cysteine protease activity. H^+^ enters the resorption lacuna and functions by dissolving bone minerals [43].

In addition, SLC4A2 is located basolaterally on the human airway epithelium and contributes to the transmembrane flow of Cl^−^ and HCO_3_^−^ in the airway epitheliums [45]. It is postulated that SLC4A2 activity can be affected by CaM under resting conditions. Recombinant Keratin 2 (CK2)-dependent phosphorylation of SLC4A2 is significant for its activity as well [39]. Furthermore, as it is expressed in the kidney, SLC4A2 is considered the main regulator that mediates HCO_3_^−^ resorption in the thick ascending limb (TAL) [40]. Additionally, SLC4A2 is evidenced to be expressed in HaCaT keratinocytes [46], which is involved in migratory dynamics [47]. Through histamine or Ca^2+^-induced stimulation, keratinocyte migration through SLC4A2 activity has been shown to be facilitated [46].

SLC4A3 is expressed most in the heart [48] and exhibits the outflow of HCO_3_^−^. Thus, it is important for the recovery of myocardial pH under an alkaline load [49], mechanical stress sensing and mechanical transduction of the heart [50]. In the brain, there are rapid and significant pH changes generated by neuronal activity, and SLC4A3 contributes to maintaining the normal function of neuronal and glial cells by expelling excess intracellular HCO3- [51]. In the early stage of metabolic acidosis (Mac), it played a primary role in the rapid decrease of pH in nervous cells. Paradoxically, SLC4A3 was very important in preventing further pH decline during the Mac period. Although it is verified that astrocytes do not express SLC4A3 [52], the data has shown that its activity is noticeable as well [53]. To some extent, the presence of SLC4A3 in neurons transmits a message to astrocytes, altering the regulation of pH. This can stimulate SLC4A4 and inhibit NHE1 [53]. As the transmembrane Cl^−^ gradient can determine the polarity and intensity of the GABAergic current [54], SLC4A3 regulates Cl^−^ activity and intracellular Cl^−^ level responses at the neurotransmitter receptors of GABA and glycine [55].

### 4.2. Sodium-Coupled SLC4 Proteins

SLC4A4 extrudes the major acid in cardiac muscle cells with the synergistic effect of NHE3 activity [56]. Through the uptake of Na^+^, it is capable of enhancing myocardial contractility through loading Ca^2+^ as the activity of sodium–calcium pump is reversed. With relatively high expression in heart [57], SLC4A4 has the ability to load Na^+^, which may impact the loading of Ca^2+^. In addition, SLC4A4 can contribute to the secretion of a large quantity of bicarbonate, helping the resorption of bicarbonate from tubular liquid back into the blood [58]. In the proximal renal tubule, apical NHE3 transports H^+^ into the lumen and mediates Na^+^ into the proximal renal tubule cells. With the synergistic power of NHE3, SLC4A4 transports the extrusion of Na^+^ and CO_3_^2−^ species to fulfill the absorption of Na^+^ and HCO_3_^−^ into the blood. Carbonate anhydrase II (CAII) catalyzes the hydration reaction in the cytoplasm, which can enhance the rate of HCO_3_^−^ absorption [4]. Expressed in the dental epithelium, SLC4A4 assists ameloblasts to secrete HCO_3_^−^ and reacts with H^+^, acting as a buffer. During the two stages of enamel development, a large quantity of H^+^ remains neutralized to fulfill a tight pH regulation [59]. Furthermore, the modulation of SLC4A4 activity can reduce the increased corneal pH and make it return to normal, as a normal eyelid opening leads to losing carbon dioxide temporarily, rapid alkaline, and an increased pH of the anterior corneal tear coat [6]. The transport process of SLC4A4 is shown in Figure 3.

In isolated connecting tubules (CNT) and cortical collecting ducts (CCD), SLC4A5 transports Na^+^ and HCO_3_^−^ outwardly from either the basolateral or luminal membrane [60]. In human iPSC-derived RPE cells, SLC4A5 was most located on the apical and basal membranes of the Golgi apparatus [10]. It is postulated that SLC4A5 plays a role in the Golgi apparatus, presumably by mediating cellular exchange, and controlling the distribution of other proteins that impact RPE ion and fluid carriage [60].

SLC4A7 is abundant in the nervous system [61], cardiac cells, and renal cells [62], and functions significantly in cerebral development [63]. SLC4A7 can accurately regulate nervous excitability in synapses [64]. The action potential, which occurs in astrocytes, can lead to the reduction of pH and the acidification of pH. The Synaptic location of SLC4A7 indicates its role in neuronal modulation, thus altering the pre- or postsynaptic pH. Located on endotheliocytes, SLC4A7 may impact the generation of endothelial NO synthase (eNOS), largely resulting from the impact of subsequently reduced pHi on eNOS [65]. Additionally, SLC4A7 can be found in the smooth muscle and blood vessel [66], where it is responsible for maintaining vasomotor responsiveness [67] and important for arterial structure [68]. SLC4A7 is expressed in the smooth muscle cells of the digestive tract, acts as a protective factor and can protect mucous membranes against the ingestion of gastric acid [66]. But its expression in cardiomyocytes remains controversial. SLC4A7 is located on the cell surface of the phagosome. The inflow of HCO_3_^−^ mediated by SLC4A7 plays a significant role in maintaining acid-base equilibrium and effective acidification of phagosome [69]. It is expressed abundantly in the parotid and submandibular glands (SMG) and mediates the influx of HCO_3_^−^ to buffer inward acidification. This indicates that SLC4A7 is essential to maintaining oral health by producing saliva with a normal concentration of HCO_3_^−^ and neutralizing the acids that are generated by bacteria in the mouse and harmful to teeth, thus reducing the risk of oral infections [70]. It is proposed that it may mediate a significant mechanism from the perspective of molecular mechanisms for secreting fluid in parotid cells. SLC4A7 participates in maintaining locomotor activity, exploratory behavior, hearing, and vision function. The exploratory behaviors are involved in sensory perception to conceive a spatial imagination in the brain. Recent research indicated that SLC4A7 plays a role in meeting nucleotide demand and is associated with cellular growth and tumor proliferation [49].

A cohort study shows that SLC4A10 is related to increased blood plasma concentrations of age-dependent interleukin IL6 [71]. Human genetic study reveals its association with the malfunction of plasma osmolality and systemic water balance [72]. SLC4A10 expression is essential to human cognitive function and nervous excitability [9].

### 4.3. The Other SLC4 Proteins

SLC4A9 is expressed on the basolateral membrane of renal β-intercalated cells [73] and in submandibular acinar cells [74]. The forkhead transcription factor Foxi1 has an activating effect on SLC4A9 [75]. SLC4A9 is important for the uptake of Cl^−^ across the basolateral membrane of acinar cells in the submandibular gland (SMG). It participates in secreting fluid dependent on the cAMP pathway, as absorbing chloride across the basement membrane is required for chloride-dependent fluid secretion [74]. SLC4A9 may account for the main mechanisms of saliva secretion and secondary Cl^−^ absorption [76]. Cooperating with pendrin and SLC4A8, SLC4A9 can contribute to absorbing NaCl in CCD and function in maintaining fluid homeostasis. With the cooperation of pendrin and SLC4A8, SLC4A9 can contribute to salt absorption in the CCD and function in fluid equilibrium and blood pressure [48] (Figure 3).

SLC4A11 can efficiently serve a function in nitrogen homeostasis and ammonia detoxification in a variety of tissues and cells [77]. Expressed on the inner membrane of the mitochondrion, it plays a role in glutamine catabolism. In the mitochondrion, glutaminolysis can produce two molecules of ammonia. This can accelerate the TCA cycle, produce NADH, and drive the electron transport chain. In the meantime, the consumption of O_2_ is increased, and hyperpolarization is facilitated. Currently, the ion-transporting mechanism of SLC4A11 remains unclear. Whether SLC4A11 mediates pH-sensitive H^+^/OH^−^ conductance or NH_3_/H^+^ model remains suspected [77]. When SLC4A11 is activated by NH_3_, the mitochondrial membrane potential (MMP) is depolarized. This can lead to an influx of H^+^ into the matrix. By fulfilling ammonia-sensitive H^+^ uncoupling, SLC4A11 can inhibit the production of mitochondrial superoxide [78] and prevent apoptosis of corneal endothelial cells (CEC) [77]. The roles of SLC4A11 in the mitochondrion are demonstrated in Figure 2. It is also known to be an oxidative stress-responsive protein [77], which is essential to proper reduced nuclear factor erythroid-related factor-2 (NRF2) activation [79]. In corneal endothelium, SLC4A11 functions as an ion pump permeating sodium, H^+^/OH^+^, ammonia, and water [80]. Therefore, it is responsible for mediating corneal transport and keeping the stroma dehydrated. Located on the basolateral membrane of the corneal endothelium, SLC4A11 contributes to the efflux of lactate. It mediates the influx of H^+^ due to the negative membrane potential, which facilitates MCT4 to cotransport H^+^ and lactate out of endothelial cells [77]. In the meantime, the lactate diffuses through a tight junction [77]. Of note, SLC4A11 is regarded as cell adhesion molecules, which adheres CEC to the Descemet membrane (DM) [81].

## 5. Molecules Interacted with SLC4 Proteins

SLC4 proteins have different domains and protein binding sites [82], and their activities may be regulated by the binding molecules [82]. These binding molecules include enzymes, ions, and second messengers, and activate the underlying signaling pathways [83]. Understanding the structure–functional characteristics and protein–protein interaction traits helps to understand the physiological functions, diversity of proteins, and clinical potential for targeted treatment.

### 5.1. SLC4A1

The N-terminal of SLC4A1 can interact with the cytoskeleton [84]. SLC4A1 is expressed on erythrocytes and contains an anchoring point for a few proteins, such as constituent parts of the cell skeleton via ankyrin (Figure 2) [84]. The ankyrin complex includes SLC4A1, glycophorin A (GPA), and protein 4.2. SLC4A1 can recruit GPA to the erythrocyte membrane. In the meantime, its activity is stimulated. The influence of the two proteins seems mutual [85]. In addition, protein 4.2 is essential to maintaining the stability of erythrocytes and also has a stimulatory effect on SLC4A1 [86]. The cytoplasmic domain of SLC4A1 can bind with protein 4.1 and form a protein 4.1-GPC junctional complex. Similar to the ankyrin complex, the junctional complex includes other proteins such as adducin, dematin, and Rh [87]. These interactions depend on the phosphorylation state of the N-terminal domain. Protein 4.1 is postulated to alter the expression of SLC4A1, which is related to the interaction of other proteins, and the C-terminal can interact with proteins, such as glyceraldehyde-3-phosphate dehydrogenase (GAPDH) [88] and carbonic anhydrase (CA) [89]. The D902EY residues in the D902EYDE motif located on the C terminal of SLC4A1 are important for GAPDH binding [88]. GAPDH is essential to the stability of SLC4A1 in the kidney, and through the stimulation of GAPDH, SLC4A1 expression in MDCKI cells can be changed [88]. Meanwhile, SLC4A1 interacts with CA II through the “DADD” motif in the C-terminal [89]. It also has a binding site for nephrin within the C-terminal, which is required for proper expression in glomeruli [83].

### 5.2. SLC4A4

Three transcripts have been identified in human *SLC4A4*: SLC4A4-A (known as kNBC), SLC4A4-B (sometimes regarded as pNBC), and SLC4A4-C (termed as hNBC). The difference between SLC4A4-A and SLC4A4-B mainly lies in their N-terminus, whereas SLC4A4-C shares an identical gene structure with SLC4A4-B except for the C-terminus [90]. Functionally, they can mediate ion transport. However, their intrinsic activity and regulation are various [91]. SLC4A4-A moves sodium and bicarbonate from epitheliums to the interstitial space in a proportion of one to three (1 Na^+^; 1 HCO_3_^−^; 1 CO_3_^2−^) [13]. SLC4A4-B transports sodium and bicarbonate from the interstitial space to epitheliums in a proportion of one to two [92].

SLC4A4-A has an autostimulatory domain (ASD) located on its N terminus [83]. This ASD is postulated to play a role in facilitating the transport rate of bicarbonate mediated by SLC4A4-A in the proximal tubule, thus promoting the renal efficiency of bicarbonate resorption. SLC4A4-B and SLC4A4-C have an autoinhibitory domain (AID) within their N-terminal [83].

IRBIT (inositol 1,4,5-trisphosphate) can bind to the inositol triphosphate (IP3) receptor, which can regulate the activity of the receptor and has a competitive effect on IP3 [93]. With the increasing concentration of IP3, IRBIT can be released from the IP3 receptor. This can have an inhibitory effect on SLC4A4 [94]. Additionally, IRBIT recruit protein phosphatase 1 (PP1) via its PP1 binding site to SLC4A4. PP1 complexes with IRBIT and makes SLC4A4 dephosphorylate, restoring its expression into cell surface. The with-no-lysine (WNK) kinases and Ste20-related proline alanine-rich kinase (SPAK) pathways can have an inhibitory effect on SLC4A4 activity [94,95]. The WNK kinases recruit SPAK into SLC4A4, and SPAK phosphorylates SLC4A4, thus stabilizing AID and exhibiting the effect of inhibition [94]. The dephosphorylation of proteins by IRBIT can block the effect of the WNK/SPAK kinases to stabilize the expression of SLC4A4 at the plasma membrane [94]. CA IV could bind to G767 on the fourth extracellular loop of SLC4A4-B. CA IV could activate the activity of all *SLC4A4* splice variants. However, it remains unclear whether CA II can directly bind to SLC4A4 [83]. Cl^−^ is a signaling molecule that regulates SLC4A4-A and SLC4A4-B. A high concentration of Cl^−^ inhibits SLC4A4-B [83]. cAMP could inhibit the absorption of HCO_3_^−^ in the kidney and promote the secretion of HCO_3_^−^ in the pancreas. SLC4A4-B activated by cAMP requires the N-terminal PKA phosphorylation site T49. Mg^2+^ can inhibit the activity of SLC4A4-B, as well. The hsp70-like stress 70 protein chaperone is able to interact with SLC4A4-B, thus increasing protein expression [6].

### 5.3. SLC4A7

SLC4A7 contains a PDZ-binding domain (ETSL motif) at the C-terminal and can interact with post-synaptic density protein 95 (PSD-95), which is expressed at the postsynaptic neuron [96]. Syntrophin γ2, which contains PDZ domains, can interact with SLC4A7 as well [97].

IRBIT can upregulate SLC4A7 activity, which was arginine-related and involved the protein phosphatase-1 and WNK/SPAK signaling pathways with an inhibitory effect [83]. Calcineurin can bind SLC4A7 through cassette II. Calcineurin has a stimulatory effect on SLC4A4-B, which is essential to avoiding smooth muscle acidosis while the artery is contracted. Additionally, vasopressin is reported to activate SLC4A7, which is located on smooth muscle arteries. SLC4A7 can bind to CAII, which can increase the transport rate [35].

### 5.4. SLC4A8

Human *SLC4A8* includes five variants, namely *SLC4A8-A-E*. It also produces a variant regarded as “THYMU3021755” without a Ct domain. The variants differ in their N-terminals and C-terminals [98]. SLC4A8 is inhibited by 4,4′-diisothiocyanato-stilbene-2,2′-disulfonic acid (DIDS) through a DIDS-recognized motif located in the fifth transmembrane domain consensus (KXXK). IRBIT was able to bind to SLC4A8-B but not SLC4A8-D, as SLC4A8-D does not include RRR sequences capable of binding to IRBIT [83].

### 5.5. SLC4A10

In mammals, the *SLC4A10* gene contains three alternative promoters and seven cassette exons. Currently, 15 “full-length” splicing variants of *SLC4A10* (SLC4A10-A-N, plus rb3NCBE) and a specific variant rb7NCBE with the isolated Nt structure have been identified [13]. IRBIT can stimulate SLC4A10, as well. In addition, the ezrin-binding protein 50 (EBP50) is able to interact with SLC4A10 [99].

## 6. Associated Diseases and Potential Clinical Values of SLC4 Proteins in Human Tissues

### 6.1. SLC4A1-3

#### 6.1.1. SLC4A1

The mutations in *SLC4A1* result in hereditary spherocytosis (HS) or South East Asian ovalocytosis (SAO) [7]. Patients with HS have manifestations such as anemia and jaundice [100]. The disruption of erythrocyte deformability in patients with HS is the main pathological mechanism of HS. The reported mutation sites of *SLC4A1* appear mostly in exons.

Mutations in *SLC4A1* also can lead to dRTA [101]. In the absence of SLC4A1 activity, HCO_3_^−^ and Cl^−^ will not be transported via SLC4A1, which leads to reduced HCO_3_^−^ concentration in the renal interstitium and retained Cl^−^ in the renal tubule [102]. Therefore, patients with dRTA are characterized by hyperchloremic metabolic acidosis, accompanied by hypokalemia. Patients during adolescence are manifested by delayed growth, rickets, kidney stones, and calcium deposits [103]. The disease is inherited through the pattern of autosomal dominant (AD) and autosomal recessive (AR) [104]. Patients with AD dRTA have no disease symptoms until adolescence or adulthood, whereas patients with AR dRTA experience severe symptoms of disease that initially develop during childhood [104]. Clinical manifestations and laboratory tests, such as the short ammonium chloride loading test, can provide the initial evidence for the diagnosis of dRTA. Furthermore, the diagnosis can be validated by genetic analysis [102]. The high-resolution melting (HRM) method can be used as molecular diagnostic tool for AR dRTA associated with *SLC4A1* mutations [104]. The effectiveness of HRM has the advantages of 100% convenience and rapidness in screening DNA specimens with the *SLC4A1* mutation. Nevertheless, HRM has not been applied to the diagnosis of the *SLC4A1* mutation in DNA specimens. In addition, PCR-RFLP and direct DNA sequencing are the conventional approaches for the analysis of the disease mutated by *SLC4A1* and suggested to be used for confirming suspicious cases [104]. Clinical therapy for dRTA includes correcting metabolic acidosis and avoiding complications. Due to the reduced concentration of potassium ions, potassium-containing preparations should be considered [5]. As dRTA can lead to nephrocalcinosis, the therapeutic strategy of replacing the kidney is essential if the disease develops to chronic renal disease and can progress to end-stage renal disease [102].

The changes in the balance of tyrosine kinase and phosphatase activities could alter the expression of SLC4A1 [83]. Oxidative stress (OS) can induce tyrosine phosphorylation of SLC4A1 and cause membrane destabilization of red blood cells. It can also impact the binding of SLC4A1 and spectrin and actin via the ankyrin bridge, as well as the interaction of SLC4A1 and hemoglobin. OS can activate the posttranslational modification in the Nt of SLC4A1, inhibit the combination of the spectrin–actin cellular skeleton, and increase the deformability of erythrocytes [8]. The impact that ROS have on erythrocytes is shown in Figure 3. Since tyr-P levels of SLC4A1 are normally stable but changes under the influence of OS-related diseases, SLC4A1 tyr-P levels can be applied to analyze the functional state of red blood cells [105]. It is notable that SLC4A1 modifications associated with OS participate in the pathology of aging [106], diabetes mellitus [107], and inflammatory diseases like endometriosis [105]. Due to the ubiquitous expression of SLC4A1 in the human body, such as the brain and lymphocytes, it can be regarded as a marker for post-translational modification during ageing [8].

Meniere’s disease (MD) is an inner ear disorder, and its pathophysiology is characterized by endolymphatic hydrops. The diagnosis of the disease is based on clinical history and examination result [108]. SLC4A1 has been demonstrated to downregulate significantly in MD and acts as an important protective factor for the disease [109]. Nevertheless, the mechanism remains unclear.

Unexplained recurrent pregnancy loss (URPL) occurs during pregnancy and is accompanied by immune dysfunction. The disease involves a complex network of cytokines. CXCL-8 can regulate the release of inflammatory cytokines. The NF-κB signaling pathway is important for the release of cytokines in the downstream signaling pathways of inflammatory factors as well. Lnc-SLC4A1-1 can interplay with NF-κB to mediate the upregulation of CXCL8, which initiates inflammation. This process may lead to apoptosis and the migration of trophoblasts, resulting in URPL [110]. This provides the possibility of finding new markers for the diagnosis and treatment of this disease. Furthermore, the Lnc-SLC4A1-1/H3K27ac/NF-κB pathway may be an underlying mechanism to mediate the anti-tumor effect of physcion 8-O-β-glucopyranoside treatment. Therefore, this signal pathway may be a possible strategy for treating endometrial cancer [111].

Acute lung injury (ALI) is a disease characterized by tissue damage leading to pulmonary epithelial dysfunction and macrophage activation [112]. Activation of the NF-κB signaling pathway leads to the development of ALI. Inhibition of this signaling pathway has clinical value for the treatment of ALI [112]. MiR-1249–5p can improve ALI by targeting and downregulating SLC4A1, which is postulated to inhibit the NF-κB signaling pathway [112].

Patients with pulmonary arterial hypertension (PAH) manifest dyspnea on exertion, fatigue, edema, palpitation, and dizziness. Vascular remodeling can increase resistance in pulmonary vessels, which leads to the increase of pulmonary artery pressure. The diagnosis of pulmonary hypertension requires right heart catheterization to assess hemodynamic status. Furthermore, an ultrasound electrocardiogram is a good test item. SLC4A1 can be a biomarker of this disease, which can represent as a higher activation of some immune cells in their peripheral blood [113].

#### 6.1.2. SLC4A2

The dysfunction of SLC4A2 participates in the pathogenesis of primary biliary cholangitis (PBC) that is a chronic, biliary obstructive, and autoimmune disease. SLC4A2 deficiency can change the pH balance of immune cells, resulting in immune disorders in PBC patients [114]. The promoter region of the *SLC4A2* gene is found to be highly methylated in the peripheral blood mononuclear cells of PBC patients. Methylation of CpG cytosines prevents DNA-binding proteins, resulting in transcriptional inactivation. Therefore, the mRNA produced by *SLC4A2* is reduced in liver and lymphocytes of patients [114]. This process leads to the downregulated expression of SLC4A2 [115]. As the expression of SLC4A2 is reduced, the biliary HCO_3_^−^ umbrella is broken. Bile salts enter cholangiocytes, which facilitate the ROS production and induce inflammation in primary biliary cholangitis [116]. The deficiency of SLC4A2 in PBC patients would make biliary cells more immunogenic and vulnerable to autoimmune injury, leading to immune disorders [115]. Chronic elevation of aspartate aminotransferase, alanine aminotransferase, alkaline phosphatase, and total bilirubin with or without specific PBC clinical manifestations of pruritus and fatigue should be suspected of PBC [114]. If the disease is not treated timely, quite a few complications, such as liver failure and death, can be caused [117]. Ursodeoxycholic acid (UDCA) has been widely demonstrated to improve the clinical outcomes of PBC [118]. The drug could restore the expression and level of SLC4A2. Through the interaction of hepatocyte nuclear factor 1 with the glucocorticoid receptor, the combination of UDCA with glucocorticoids is able to activate the promoter of SLC4A2 in human hepatocytes [114]. MiR-506 can bind to the 3′ untranslated region of *SLC4A2* mRNA, prevent the translation of mRNA into protein, and impair the function of bicarbonate secretion in the biliary tract. MiR-506 is up-regulated in the biliary tract cells of PBC patients and is considered to be a therapeutic target for PBC [119].

Mutations in *SLC4A2* can cause the occurrence of osteopetrosis [120]. The *SLC4A2* mutation can affect the differentiation of osteoclasts [41]. As the differentiation process of osteoclasts is affected, the mineralized material of the bone matrix fails to dissolve, and bone mineral density can be increased. This can lead to the occurrence of osteogenesis. Mutations in *SLC4A2* affect cysteine protease activity, leading to the formation of abnormal podosome bands [42]. The aberrant podosome bands can impair bone resorption, which leads to an osteoporotic phenotype. The dysfunction of SLC4A2 would also break the dynamic organization of osteoclasts to maintain acid-base balance [42] as well. Patients with osteopetrosis have fractures and stunted growth. The diagnosis of osteogenesis is mainly based on a bone imaging examination. In the absence of imaging studies, elevated concentrations of creatine kinase, BB isoenzymes, and tartrate-resistant acid phosphatase are helpful in the diagnosis of autosomal dominant osteopetrosis (ADO). Genetic testing can be used to detect and distinguish different subtypes of osteoporosis [121]. As *SLC4A2* plays a role in forming the actin rings of osteoclasts, it is considered a pathogenic gene to treat periprosthesis osteolysis. DIDS, which blocks SLC4A2 expression, has a positive effect on the reduction of the region that resorbs the bone, providing plausible evidence for its role in the therapeutic strategy of osteoclastic-associated osteolytic diseases [122].

The expression of SLC4A2 was found to be regulated in quite a few cancers. It was upregulated in the esophageal squamous cell carcinoma (ESCC), hepatocellular carcinoma [123], and colon tumor tissue, but downregulated in the gastric cancer (GC) cells [124]. In ESCC, the decreased SLC4A2 expression facilitates intracellular alkalinization, which promotes cancer cell metabolism [125] and is correlated with a poor prognosis [126]. Additionally, SLC4A2 was correlated with the proliferation [127] and migration [128] of the hepatoma cell. However, whether SLC4A2 affects the other biological behaviors of liver cancer requires further research. As the inhibition of SLC4A2 expression can decrease the proliferation of cancer cells, it may have a function in promoting colorectal cell growth. The expression of SLC4A2 can be inhibited by gastrin, leading to the inhibition of cancer proliferation [129]. p16 can bind with SLC4A1 and SLC4A2. The combination of p16 and SLC4A1 can facilitate the degradation of SLC4A2 in GC cells. Its downregulation in GC cells is partly attributed to the mediation of the ubiquitin proteasome pathway [130]. SLC4A2 is stimulated by transcription factor early growth response 1 in a cholecystokinin B receptor-dependent manner. The combination of tastuzumab and gastrin inhibits human epidermal growth factor receptor 2-negative GC cells, thereby inhibiting the complex of SLC4A1 and pl6, which may upregulate the expression of SLC4A2 in GC tissues [131]. As SLC4A2 is correlated with the poor differentiation and prognosis of the ESCC, hepatocellular carcinoma [123], gastric cells [130], and colon cancer [129], it is considered an underlying target for diagnosing and treating these diseases.

#### 6.1.3. SLC4A3

There are quite a few studies suggesting that SLC4A3 is associated with heart disease. Short QT syndrome (SQTS) is an inherited disorder caused by a defect in potassium and calcium channels that leads to an abnormally short QT interval. Recently, it was reported that the mutation in *SLC4A3* causes SQTS [132]. Inhibited activity of SLC4A3 can lead to increased intracellular pH and decreased concentration of intracellular Cl^−^. This can affect the activity of other channels expressed on the myocytes. For example, Kv7.1 channels (KCNQ1) are activated and L-type Ca^2+^ channels are inhibited. The process can contribute to repolarization. Furthermore, the reduction of intracellular chloride concentration can make its equilibrium potential more negative, thus increasing the inflow of chloride in the second stage and at the start of the third stage and reducing the time of action potential. In addition, chloride channels in cardiomyocytes are inhibited due to the decrease in intracellular Cl^−^ concentration. This may directly lead to the arrhythmia in SQTS. The most common clinical manifestation of SQTS is cardiac arrest. Other clinical manifestations include palpitations and syncope. An implantable cardioverter defibrillator is the first-line treatment for SQTS. Quinidine can prolong the QT interval and is effective in the treatment of SQTS, especially in those patients who have contraindications to defibrillators and are rejected [133]. Currently, the role of genetic testing in the diagnosis of SQTS has not been elucidated. It has been demonstrated that *SLC4A3* should be incorporated into the genetic screening of patients with SQTS [134].

Catecholaminergic polymorphic ventricular tachycardia (CPVT) is a rare inherited arrhythmia disease that can cause sudden cardiac death. Similar to SQTS, genetic screening is widely used for this rare genetic disorder. SLC4A3 has an antidiastole value for diagnosing CPVT [135] as well.

In addition, the deficiency of SLC4A3 can lead to rapid decompensation, make heart failure occur more easily [136], and disturb normal cardiac function to react efficiently to acute stress [137].

Idiopathic generalized epilepsy (IGE) is an age-related, recurrent, generalized seizure with no obvious trigger, no detectable brain damage, and no metabolic disorder. Genetic factors play an important role in the etiology of IGE [51].

In the brain, the Ala867Asp variant in *SLC4A3* is associated with epilepsy [51]. The variants are evidenced to lead to decreased SLC4A3 transport activity, resulting in abnormal intracellular pH and cell volume changes, which may facilitate neuronal hyperexcitability and seizures [138]. But the molecular basis for this effect has not been determined [139]. Patients with SLC4A3 deficiency present with the retinal pathological phenotype of most vitreoretinal degeneration [55].

Early stages of non-small cell lung cancer (NSCLC) can be classified by surgical pathology. The research indicates that detection of SLC4A1 and SLC4A3 can predict the prognosis in patients with the early stage of NSCLC [140]. Cells are capable of osmoregulation by activating a number of transporters. During cell transformation, ion channels become dysregulated. As SLC4A3 is involved in cellular transformation, it can be a good marker and an excellent therapeutic target for transformative diseases such as cancers [141].

### 6.2. Sodium-Coupled SLC4 Proteins

#### 6.2.1. SLC4A4

Recessive mutations in *SLC4A4-A* lead to proximal renal tubule acidosis (pRTA) [6]. pRTA is an inherited disorder that is characterized by reduced HCO_3_^−^ resorption in the proximal renal tubule. Due to the defect of HCO_3_^−^ resorption, pRTA represents metabolic acidosis such as hypokalemia, normal anion-gap metabolic acidosis, and aciduria (pH < 5.5) [142]. As SLC4A4-A is expressed in other tissues like the nervous system, pRTA can also present with extrarenal manifestations such as developmental and intellectual disabilities, ocular defects such as cataracts and glaucoma, and dental defects [143].

SLC4A4-B can be linked to a primary headache, largely due to the dysregulation of brain local pH. To some extent, the reduction in SLC4A4-B activity may cause hemiplegic migraine in astrocytes, which is accompanied by a complex aura that includes a motor defect. Misfolded SLC4A4-B can cause abnormal NMD-mediated neuronal hyperactivity, which may be the pathogenesis of migraine in homozygotes [144].

*SLC4A4* is located on chromosome 4 and indicates that it may be related to dental development. Additionally, it is also involved in the pH regulation of amelogenesis. *SLC4A4* can be a candidate gene for amelogenesis imperfecta in human diseases [145].

Furthermore, SLC4A4 participates in the development of type 2 diabetes mellitus (T2DM), which is due to perturbation of the β cell’s transcriptional regulation. The abnormal regulation of glucose-stimulated insulin secretion (GSIS) is partly assigned to the mitochondrial dysfunction, which is a major constituent of β cell failure in T2DM. The increased SLC4A4 activity is associated with intracellular alkalinization in tumoral β cells [146]. The upregulated expression of SLC4A4 can lead to intracellular alkalinization and impair mitochondrial function, which contributes to β cell functional failure and glucose intolerance. Through inhibiting SLC4A4 with S0859, pH_i_ can be lowered and GSIS is enhanced in T2DM human islets. Thus, inhibition of SLC4A4 may be considered a potential strategy to counteract β cell failure in T2DM [147].

SLC4A4 participates in acute and chronic hypoxia along with ischemia [148] in neurons and glia, such as the interrupted supply of some significant substances like oxygen and glucose, as well as the impeded synthesis of ATP. These metabolic processes can eventually lead to extracellular acidosis. As the intracellular and extracellular environment is acidic during ischemia, SLC4A4 uses the electric potential of Na^+^ to fulfill the transport of Na^+^ and HCO_3_^−^, thus regulating pH. During the reperfusion, while extracellular pH returns to normal and intracellular pH remains acidic, the pH gradient helps H^+^ extrusion [149]. NHEs, by means of extruding H^+^, provide the way for the entry of HCO_3_^−^ through NBCs. As ATP synthesis is prevented and the activity of Na^+^/K^+^ ATPase decreases, the intracellular concentration of Na^+^ is increased, which induces cell depolarization and stimulates the abnormal release of excitatory amino acid transmitters. Due to the reversed transport of the Na^+^/Ca^2+^ exchanger, the intracellular concentration of Ca^2+^ increases and promotes a variety of calcium-activated cell damage processes [150].

Prostate cancer (PC) is a common disease that can affect men’s health, and its diagnosis and treatment are complex [151]. As circle RNA is found to have potential value for the diagnosis and treatment of diseases, hsa_circRNA_001587 is applied to the research. The experiment has indicated that increased activity of hsa_circRNA_001587 can upregulate the expression of SLC4A4, which curbs the neoplastic processes of PC through binding to miR-223. The hsa_circRNA_001587-miR-223-SLC4A4 axis plays a role in the development of PC [152]. Nevertheless, the tumorigenic mechanisms of PC require further research. Through the regulation of the AKT pathway, SLC4A4 can promote the progression of PC, and its inhibition can be an excellent therapeutic strategy for treating the disease [153].

SLC4A4 is evidenced to be the most abundant HCO_3_^−^ transporter expressed in pancreatic ductal adenocarcinoma (PDAC) [154]. The low pH of the tumor microenvironment can lead to aberrant function of immune cells such as CD8+T cells and impact the efficacy of immune checkpoint inhibitors. Inhibition of SLC4A4 activity can increase the accumulation of bicarbonate in the extracellular space and reduce the secretion of lactate, thereby alleviating acidosis in the acidic tumor microenvironment. Inhibiting SLC4A4 is considered a possible remedy to improve immunoreaction, which inhibits tumor growth and metabolic processes. SLC4A4 can be a therapeutic target to tackle immunotherapy resistance and prolong survival in PDAC [154].

As the reduced expression of *SLC4A4* in patients with colon adenocarcinoma is associated with lymph node invasion and distant metastasis, this gene is supposed to be a biomarker to predict the poor prognosis of patients [155].

#### 6.2.2. SLC4A5

According to transcriptome analysis, it is hypothesized that SLC4A5 activates a regulatory cascade and is composed of compensated HCO_3_^−^ reuptake through other transporters that mediate the transport of Na^+^ and HCO_3_^−^ (e.g., SLC4A7), thus causing increased Na^+^ absorption. This will increase blood pressure and lead to hypoaldosteronism, which explains the connection of the SLC4A5 locus to hypertension in humans from the perspective of molecular mechanism [156]. According to some studies, HCO_3_^−^ transport mediated by SLC4A5 in the choroid plexus epithelium (CPE) is the major molecular mechanism to regulate the cerebrospinal fluid (CSF) during respiratory acidosis [157]. *SLC4A5* is downregulated in Alzheimer’s disease and considered to be a candidate gene to produce CSF in AD. The altered expression of SLC4A5 can adversely impact the normal function of CSF secretion by impacting the carriage of electrolytes and water from CPE to CSF [156,158].

#### 6.2.3. SLC4A7

Genome-wide association studies discovered that a variant of *SLC4A7* is associated with blood pressure [159]. SLC4A7 is related to hypertension due to vascular change. Mutations in *SLC4A7* lead to mildly reduced blood pressure on the account of the altered vessels and the inhibition of NO synthase and Rho kinase [160]. Reduced endothelial NO production can result in downregulated arterial dilatation and mildly upregulated blood pressure. In addition, the loss of SLC4A7 activity causes reduced intracellular pH, thereby influencing local signals that regulate arterial dilatation and arterioconstriction. The depression of the Rho kinase signal channel is regarded as the potential mechanism of the changed vascular function. SLC4A7 may represent a new target, and its inhibition provides a new approach for treating cardiovascular disease.

The research on alcohol addiction demonstrated that the defective activity of SLC4A7 can elevate the consumption of alcohol and make the body more susceptible to sedation induced by alcohol. The underlying mechanism may be intracellular acidosis and decreased nerve excitability. Chronic alcohol consumption in mice reduces the expression of *Slc4a7* in a positive feedback manner, suggesting that *Slc4a7* plays an important role in regulating alcohol consumption and susceptibility to alcohol-induced sedation [161].

In addition, chronic acidosis is related to increased poisonousness mediated by glutamate, causing neurologic impairment. Acidosis may cause ATP consumption and a depolarized membrane, which eliminates Mg^2+^. N-methyl-D-aspartate receptors are known to be activated to trigger apoptosis and lead to cytotoxicity under no or decreased Mg^2+^ concentrations. Therefore, neuronal death is further promoted to some extent [162]. Due to its significant function in neural damage, SLC4A7 may be considered a new neuroprotective target for brain damage induced by glutamate [163]. Mutation in *SLC4A7* can lead to reduced locomotor activity, which may result from an alteration in exploratory behaviors or emotional ability. Furthermore, deficits in visual and acoustic faculty can impact affective and cognitive function [11]. The altered perception of sensory cues may impact animals’ capability to explore their environments for survival and adaptation. However, it has been studied that SLC4A7 displays few roles in motor ability [164].

SLC4A7 is revealed to upregulate in the carcinoma cell line in the presence of the MCF-7 Nt-truncated ErbB2 receptor (NErbB2). This overexpression increases carcinomas’ acid excretion ability and alleviates the acid load inside cells generated from glycolysis, thus regulating intracellular pH. It is postulated that *SLC4A7* can impact breast carcinoma by acting as a modulator or a tyrosine kinase substrate through the development of carcinomas [165]. Furthermore, SLC4A7 can impact the progression of head and neck squamous cell carcinoma (HNSCC). Through the activation of the PI3K/AKT/mTOR signaling pathway, SLC4A7 contributes to the migration and invasion of HNSCC. The role of SLC4A7 in the PI3K/AKT/mTOR signaling pathway indicates that it can act as a predictive biomarker and therapeutic target in HNSCC [166].

#### 6.2.4. SLC4A10

The disruption of SLC4A10 can lead to epilepsy [9]. Due to the alteration of acid-base equilibrium, neuronal excitability is impacted. This can make epileptic phenotypes easier to develop. Additionally, SLC4A10 is hypothesized to function in the pathology of primary open-angle glaucoma (POAG). The clinical manifestation of POAG is chronic optic lesion and progressive loss of retinal ganglion cells, leading to specific visual field defects [167]. SLC4A10 participates in CSF production and can impact translaminar pressure [167]. A decrease in CSF pressure and increased Intraocular Pressure can cause an increased translaminar pressure difference, thus disturbing axoplasmic flow and induced retinal apoptosis.

#### 6.2.5. SLC4A8

Currently, it remains unclear whether SLC4A8 plays a role in the development of salt-dependent hypertension. Research indicates that SLC4A8 and SLC26A4 mediate the reabsorption of electrically neutral sodium chloride in renal cortical collecting ducts. This provides insights into the treatment of arterial hypertension and contributes to the understanding of the regulation of sodium and potassium homeostasis by cortical collecting ducts.

### 6.3. The Other SLC4 Proteins

Mutations in the *SLC4A11* gene get involved in corneal dystrophies, such as congenital hereditary endothelial dystrophy (CHED) [168], which is a rare corneal endothelial dysfunction. The most common cause of CHED is misfolded proteins that prevent the protein from maturing and trafficking to the plasma membrane. Other causes include oxidative stress due to misfolded proteins and compensatory changes in other gene products [169]. The endoplasmic reticulum (ER) is an organelle that recognizes misfolded proteins. When the formation rate of misfolded proteins reaches saturation, it can lead to ER stress [170]. ER stress occurs in CHED [171]. In addition, SLC4A11 was able to adhere to the descemet’s membrane. When this adhesion function is lost, it leads to the occurrence of CHED and Fuchs’ endothelial corneal dystrophy (FECD) [81]. FECD is a progressive, overt disease with the onset of symptoms in the 40th to 50th years of life. FECD can be affected by mutations in other genes, while CHED is only caused by mutations in *SLC4A11* [81]. Due to its pathogenesis, it is proposed that correcting misfolding is an effective therapeutic strategy. Non-steroidal anti-inflammatory drugs can effectively treat some hereditary FECDs whose etiology has been identified by genetic testing [172]. Furthermore, a dysfunction of SLC4A11 can lead to the generation of mitochondrial ROS, which can damage the mitochondria and promote mitophagy. In the meantime, the function of lysosomes is destroyed and aberrant [171]. Research indicates that patients with CHED mutated by the homozygous *SLC4A11* can develop Harboyan syndrome at a later age [173], which can lead to progressive sensorineural hearing loss.

Glutamine is known to be required by carcinomas proliferation [174], known as “glutamine-addicted” cancers. It can provide cells with essential substrates. SLC4A11 is significantly upregulated and seems to be a risk factor for ovarian carcinoma [175]. It has been found that increased SLC4A11 expression is related to the poor prognosis of colon cancer [176]. Experimental trails demonstrate that inhibiting the activity of SLC4A11 and glutaminase (GLS1) could have a very robust effect on disrupting glutamine-addicted cells.

## 7. Homeostasis of Bicarbonate in the Body and Drugs Affecting Bicarbonate Transport

Carbon dioxide can be produced through respiration, which is able to diffuse freely through the lipid bilayer and spontaneously cross the membranes of various cells and organelles [177]. This molecule can react with water through the catalysis of carbonic anhydrase to form bicarbonate and hydrogen ions. Bicarbonate is unable to diffuse freely through the lipid bilayer and is transported across the membrane through ion channels distributed in the membrane [177]. SLC4 proteins belong to the bicarbonate transporter of the human body and are selective for substrates. The majority of SLC4 proteins were shown to be permeable to bicarbonate. Transmembrane transport of bicarbonate plays a role in many different organs and tissues, which is essential to acid-base homeostasis in the human body [178]. Bicarbonate mediates a variety of physiological functions in the human body and can regulate intracellular pH, impact membrane potential due to the formation of driving forces, maintain the excitability of neurons, and keep the normal characteristics of fluid in the human body [179]. Currently, stilbene derivatives (such as DIDS and SITS) and the N-cyanosulfonamide compound S0859 are well-studied inhibitors of sodion-coupled bicarbonate transporters [4]. Blockers, which includes non-steroid anti-inflammatory drugs, oxosol dyes, can inhibit SLC4A4 [4]. DIDS is a classical anion exchange inhibitor and is able to interact with the cell membrane to prevent cell penetration [4]. S0859 can inhibit the activity of Na^+^-coupled HCO_3_^−^ transporters with high effectiveness and specificity. It has been confirmed that S0859 exhibits an inhibitory effect on SLC4A7 activity in the MCF-7 human breast cancer cell line. SLC4A7 is able to regulate intracellular pH through the uptake of bicarbonate, which has a neutralizing effect on the metabolic acid products. Acidic wastes cannot be metabolized in a timely manner through disrupting SLC4A7, which is not conducive to the growth of tumors [180]. The use of SLC4A7 inhibitors has the potential to disrupt the growth of tumors [181]. In addition, the use of S0859 on the cardiac muscle cells deaccelerates the recovery of acidosis, which indicates the role of SLC4A7 in the heart [182].

## 8. Discussion

The SLC4 family consists of ten members. Each protein has its own structural features and distribution. When these proteins are mutated, they lead to a number of genetic diseases. Clinical manifestations and imaging diagnoses are the basis for the diagnosis of these diseases. With the development of technology, more and more molecular diagnostic methods will be applied to the genetic screening of these diseases. In addition, some SLC4 proteins can also be used for disease diagnosis. For example, the tyrosine phosphorylation status of SLC4A1 can be used as an index to evaluate the functional status of red blood cells. At present, there are some SLC4 molecules involved in the occurrence and development of diseases through signaling pathways, which also has implications for the diagnosis and treatment of diseases. For instance, lnc-SLC4A1-1/H3K27ac/NF-κB and PI3K/AKT/mTOR are signaling pathways related to SLC4 molecules. Experiments have shown that inhibitors of SLC4 proteins can prevent the development of some diseases. More studies on the mechanism of SLC4 molecules will help to understand the occurrence and development of related diseases and provide strategies for treatment.

We have described the functions and mechanisms involved in the diseases, especially the current physiological roles of SLC4 family members. Most SLC4 proteins are responsible for the excretion or secretion of HCO_3_^−^. In spite of extensive learning currently on their functions in tissues and organs, it should further comprehend the importance of SLC4 family members in terms of ion dependence, structural properties, pharmacological traits, etc. Although their molecular mechanisms and therapeutic prospects remain limited and more investigation is needed, we still expect the promising results of the SLC4 family in the short run.

## Figures and Tables

**Figure 1 ijms-24-15166-f001:**
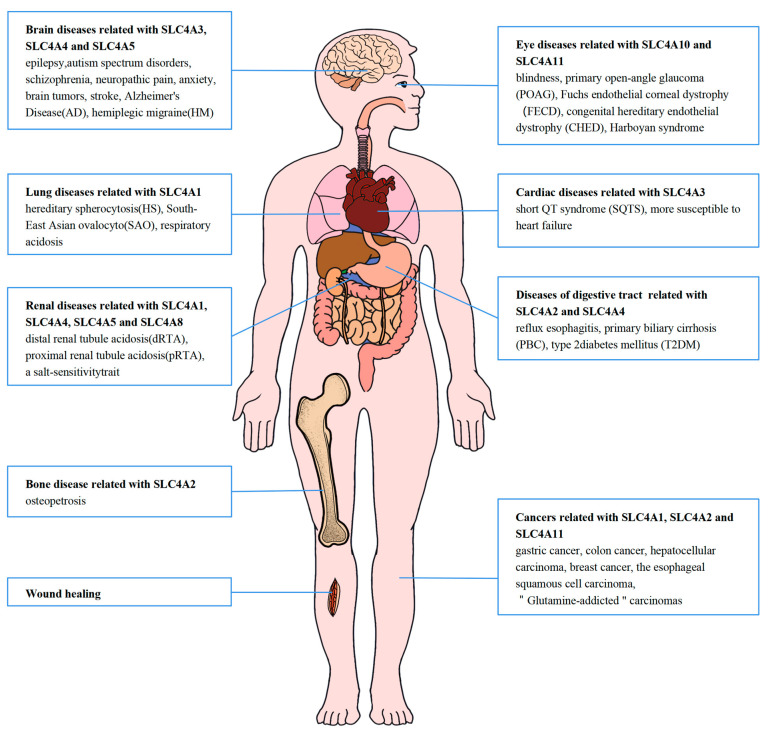
SLC4 molecular related diseases. Various SLC4 proteins are ubiquitously distributed in the human body and the mutations that these molecules may cause across the body.

**Figure 2 ijms-24-15166-f002:**
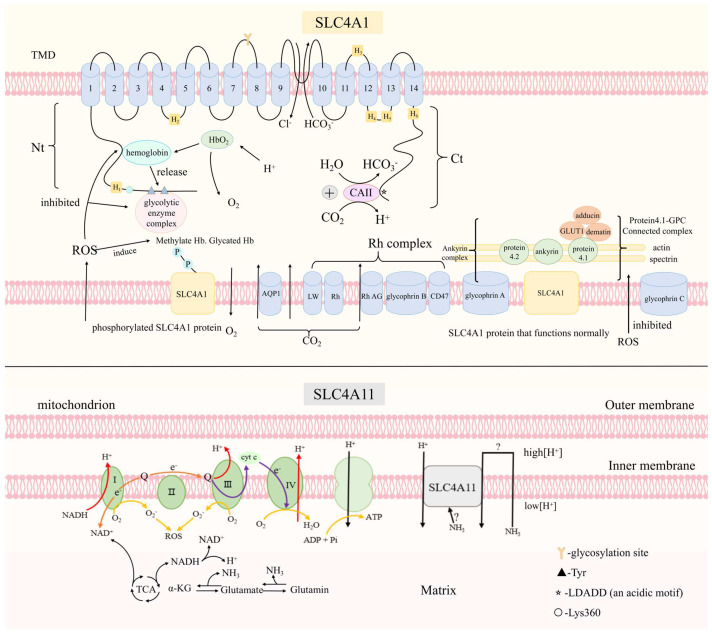
The role of SLC4A1 and SLC4A11. SLC4A1 is expressed on the erythrocytes and plays a significant role in respiratory system such as Jacobs–Stewart cycle. In the tissue space, CO_2_ enters into erythrocytes by diffusion or transport through AQP1 and Rh associated glycoprotein (RhAG). Carbonic anhydrase II catalyzes CO_2_ and H_2_O, forming HCO_3_^−^, H^+^. Acting as a buffer, hemoglobin combines with H^+^, which is companied with the release of oxygen and its diffusion to the tissues. Bicarbonate is exchanged for chloride outside the red blood cells through SLC4A1. Additionally, oxidative stress (OS) can cause erythrocytes to activate tyrosine kinases, inducing tyrosine phosphorylation which is located at the cytoplasmic domain of SLC4A1. In addition, OS can also impact the binding of SLC4A1 and spectrin, actin via the ankyrin bridge, as well as the interaction of SLC4A1 and hemoglobin. Reactive oxygen species (ROS) can induce the generation of methemoglobin and glycated hemoglobin. In mitochondrion, glutamine breaks down into glutamic acid and further yields α-ketoglutaric acid, accelerating TCA cycle. The process can lead to increased consumption of O_2_, accelerate the production of O_2_^−^ and is consistent with the increase of hyperpolarizing. The ion-transporting mechanism of SLC4A11 remains unclear. Whether SLC4A11 conducts H^+^/OH^−^ model or NH_3_/H^+^ model is still suspected by researchers. When SLC4A11 is activated by NH_3_, the mitochondrial membrane potential (MMP) is depolarized. This can lead to the influx of H^+^ into the matrix. When SLC4A11 mediates the transport of H^+^/OH^−^, its activity can be stimulated by pH.

**Figure 3 ijms-24-15166-f003:**
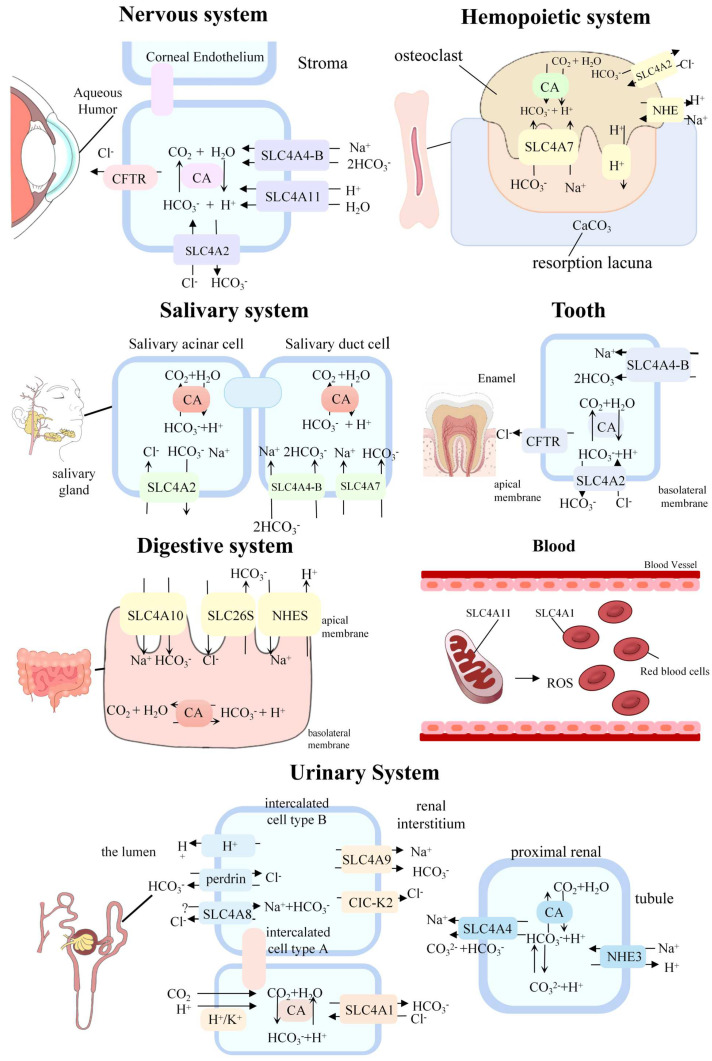
Associated physiological mechanism of SLC4 family. In the corneal endothelium, SLC4A4-B can transport one sodium and two bicarbonates. SLC4A11 functions as an ion pump permeating H^+^/OH^−^ and water. SLC4A2 can be included in the basolateral components of the endothelial pump. In osteoclasts, H^+^ secretion is supported by SLC4A2 in the contra-lacunar membrane. Liberated HCO_3_^−^ is absorbed across the lacunar membrane by SLC4A7 and across the contra-lacunar membrane by SLC4A2. In salivary glands, SLC4A4 and SLC4A7 mediate transepithelial HCO_3_^−^ secretion by salivary gland duct cells. SLC4A2 can regulate intracellular pH by transporting bicarbonates out of salivary acinar cells. In ameloblasts, SLC4A4-B assists in secreting HCO_3_^−^ and buffer protons released by mineral formation. SLC4A2 can exchange one bicarbonate for one chloride to secrete bicarbonates into the enamel space. In the small intestine, NHE3 mediates the exchange of outward Na^+^ and inward H^+^. At the same time, solute carrier family 26 (slc26) exchangers mediate the exchange of outward Cl^−^ and inward HCO_3_^−^. The apical SLC4A10 mediates direct absorption of bicarbonate and sodium into the small intestine epithelium and can be considered the equivalent of Na⁺/H⁺ exchanger 3 (NHE3) and carbonic anhydrase. As ROS is generated by neutrophils and macrophages and may be released into the blood stream, the properties and homeostasis of erythrocytes can be affected by structural and functional alteration of the transporter. In the kidney, SLC4A1 mediates the efflux of bicarbonate and the influx of chloride in α-intercalated renal tubule cells. SLC4A8 and SLC4A9 are expressed in B-type intercalated cells of the renal tubule. With the cooperation of pendrin and SLC4A8, SLC4A9 can contribute to salt absorption in the CCD. Nevertheless, the location of SLC4A8 is suspected by some researchers. In proximal renal tubule, SLC4A4-A transports the extrusion of sodium and carbonate species to fulfill the absorption of Na^+^ and bicarbonate into the blood with the synergistic power of apical NHE3.

**Table 1 ijms-24-15166-t001:** Expressions, Functions, and Pathology of SLC4 Proteins.

Protein Name	Expression Sites	Physiological Functions	Pathological Processes
SLC4A1	Erythrocytes, renal intercalated-A cells, epididymis	Participation in gas exchange, regulation of pH in the blood, involvement in sperm capacitation and rearrangement	Instability of the erythrocyte lipid bilayer (HS), changed concentration of bicarbonate and chloride in the kidney and blood (dRTA), phosphorylation of SLC4A1 and deformability of erythrocytes (oxidative stress), initiation of inflammation (unexplained recurrent pregnancy loss), activation of the NF-κB signaling pathway (acute lung injury)
SLC4A2	Esophagus, stomach, small intestine, pancreas, cholangiocytes, airway epitheliums, osteoclasts, keratinocytes	Regulation of pH in digestive tract, involvements in osteoclast differentiation, apoptosis and maturation, participation in cytoskeletal organization of osteoclasts, mediation in bicarbonate resorption of TAL, regulation of keratinocyte migration	Immune disorders and broken bicarbonate umbrella of the bile duct (primary biliary cholangitis), affected differentiation of osteoclasts and increased bone mineral density (osteopetrosis), facilitation of intracellular alkalinization and promotion of cancer cell metabolism (esophageal squamous cell carcinoma)
SLC4A3	Cardiomyocytes, neurons, glial cells	Participation in recovering pHi of myocardial cells, involvements in cardiac mechanical conduction, maintenance of pH in nervous cells, signal transmission of astrocytes, regulation of Cl^−^ at the neurotransmitter receptor	Association with some heart diseases, epilepsy
SLC4A4	Heart, proximal renal tubule, ameloblasts, corneal epithelial cells	Impact on myocardial contractility, participation in bicarbonate absorption of proximal renal tubule, secretion of bicarbonate in ameloblasts, regulation of pH in corneal epithelial cells	Defect of bicarbonate resorption (proximal renal tubule acidosis), dysregulation of brain local pH (primary headache), abnormal NMD-mediated neuronal hyperactivity (migraine), dysregulation of pH during amelogenesis (amelogenesis imperfecta), perturbation of the β cell’s transcriptional regulation (type 2 diabetes mellitus), regulation of pH (ischemia), promoting role (prostate cancer), regulation of pH and impact of efficacy in some immune cells (pancreatic ductal adenocarcinoma), association with lymph node invasion and distant metastasis (colon adenocarcinoma)
SLC4A5	Isolated connecting tubules (CNT), cortical collecting ducts (CCD), Golgi apparatus	Mediation of ion exchange on the membrane of kidney and RPE	Increased blood pressure and hypoaldosteronism(hypertension), changed CSF production (Alzheimer’s disease)
SLC4A7	Nervous system, cardiac cells, renal cells	Modulation of neurons, impact on the activity of endothelial NO synthase (eNOS), maintenance of vasomotor responsiveness and arterial structure, neutralization of gastric acid, maintenance of acidification of phagosome, production of bicarbonate in the saliva, maintenance of brain function, association with cellular growth and tumor proliferation	Modulation of neurons, impact on the activity of endothelial NO synthase, maintenance of vasomotor responsiveness and arterial structure, neutralization of gastric acid, maintenance of acidification of phagosome, production of bicarbonate in the saliva, maintenance of brain function, association with cellular growth and tumor proliferation, vascular change and the inhibition of NO synthase and rho kinase (hypertension), intracellular acidosis and decreased nerve excitability (chronic alcohol consumption and susceptibility to alcohol-induced sedation), alteration in exploratory behaviors or emotional ability, deficits in visual and acoustic faculty, the altered perception in sensory cue (reduced locomotor activity), regulation of pH (breast carcinoma), activation of PI3K/AKT/mTOR signaling pathway (head and neck squamous cell carcinoma)
SLC4A8	Brain, pituitary gland, testis, the trachea, thyroid, kidney, and pancreas	Mediation of electroneutral NaCl absorption in type B intercalated cells of CCD	Dysregulation of ion transport in renal cortical collecting ducts (salt-dependent hypertension)
SLC4A9	Renal β-intercalated cells, submandibular acinar cells	Uptake of Cl^−^ in the submandibular gland (SMG), contribution to absorption of NaCl in CCD, maintenance of fluid homeostasis	
SLC4A10	Brain	Association with plasma osmolality and systemic water balance, maintenance of brain function	Alteration of acid-base equilibrium, impact on neuronal excitability (epilepsy), alteration of CSF production, impact on the translaminar pressure (primary open-angle glaucoma)
SLC4A11	Mitochondrion, corneal endothelial cells	Function in nitrogen homeostasis and ammonia detoxification, mediation of corneal transport, adhesion action of CEC to DM	Misfolded protein, oxidative stress, endoplasmic reticulum stress, the lost function of adhesion (congenital hereditary endothelial dystrophy), the lost function of adhesion (Fuchs’ endothelial corneal dystrophy), upregulated activity (“glutamine-addicted” cancers)

## Data Availability

Not applicable.

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
