# Peer review of "Potential Theranostic Roles of SLC4 Molecules in Human Diseases"

_ijms, 2023, doi:10.3390/ijms242015166_

Round 1

Reviewer 1 Report

SLC4 carriers are responsible for transmembrane ion transports and bicarbonate secretion/ excretion in various epithelial cells. Besides ion transport, they mediate a series of other physiological functions including molecular signal transduction, pH regulation of body fluids, cell volume regulation, amd protein-protein interactions. SLC4 proteins have ten different members encoded by the SLC4A1-5 and A7-11 genes. They could be detected in various organs, such as kidney, gastrointestinal tract, brain, endocrine and hematopoietic systems. SLC4 proteins participate in a series of pathologies and are target of increasing number of clinical trials. Because of the diverse role of SLC4 transporters, reviews summarizing their physiology and pathology have a great importance.

Comments

This reviewer suggests to the Authors to add additional sections to the paper for better understanding SLC4 transporter functions. These are: Homeostasis of HCO3- ions in the body and drugs affecting bicarbonate transport.

It would help the readers to digest the biological aspects of SLC4 transporters, if more explanatory figures were added into the text or if Fig. 3. would be divided into more, separate figures.

Moreover, I would suggest to construct a table containing SLC4 members, site of their expression, their physiological functions, and if known, their involvement in various pathologies.

I would also suggest to improve figure quality as it is difficult for the readers to read in their current conditions.

Reviewer 2 Report

In the manuscript by Zhong et al. the roles of SLC4 transporters in human diseases and use as a drug target are discussed. Please consider the following comments to improve the manuscript:

1) Fig. 1. Please add schematic information about the function of transporters to the figure and the names of the proteins. 

2) Lines 150 - 169, lines 172 - 184, lines 327 -342. Please provide references.

3) There are a lot of typos in the text. English check is required. Many sentences are not clear (e.g. Line 776).

4) Please check that all abbreviations are explained. For instance, NHE, CK2, etc.

5) In section 4,  would recommend to provide a table summarizing information about expression of SLC4 transporters in different tissues and their role there.

6) In the introduction, authors state that developing of SLC4 inhibitors  should be exlpored. This should be also covered in the Discussion. Especially, as the decrease in activity of transporters can lead to several diseases as the authors discussed, how this approach will be achieved? 

There are a lot of typos in the text. English check is required. Many sentences are not clear (e.g. Line 776).
